# Association of CARD14 Single-Nucleotide Polymorphisms with Psoriasis

**DOI:** 10.3390/ijms23169336

**Published:** 2022-08-19

**Authors:** Saima Suleman, Gagan Chhabra, Rubab Raza, Arslan Hamid, Javed Anver Qureshi, Nihal Ahmad

**Affiliations:** 1Department of Dermatology, University of Wisconsin, Madison, WI 53705, USA; 2Center for Research in Molecular Medicine, Institute of Molecular Biology and Biotechnology, University of Lahore, Lahore 54000, Pakistan; 3Department of Biochemistry, Faculty of Biological Sciences, Quaid-i-Azam University, Islamabad 15320, Pakistan; 4The Life & Medical Sciences Institute (LIMES), University of Bonn, 53113 Bonn, Germany; 5William S. Middleton VA Medical Center, Madison, WI 53705, USA

**Keywords:** psoriasis, CARD14, SNPs, Sanger sequencing

## Abstract

Psoriasis is an immune-mediated chronic and painful disease characterized by red raised patches of inflamed skin that may have desquamation, silvery-white scales, itching and cracks. The susceptibility of developing psoriasis depends on multiple factors, with a complex interplay between genetic and environmental factors. Studies have suggested an association between autosomal dominant CARD14 (caspase recruitment domain-containing protein 14) gain-of-function mutations with the pathophysiology of psoriasis. In this study, non-synonymous single-nucleotide polymorphisms (nsSNPs) of CARD14 gene were assessed to determine their association with psoriasis in Pakistani population. A total of 123 subjects (63 patients with psoriasis and 60 normal controls) were included in this study. DNA was extracted from blood, and PCR analysis was performed followed by Sanger sequencing for 18 CARD14 specific nsSNPs (14 previously reported and the 4 most pathogenic nsSNPs identified using bioinformatics analysis). Among the 18 tested SNPs, only 2 nsSNP, rs2066965 (R547S) and rs34367357 (V585I), were found to be associated with psoriasis. Furthermore, rs2066965 heterozygous genotype was found to be more prevalent in patients with joint pain. Additionally, the 3D structure of CARD14 protein was predicted using alpha-fold2. NMSim web server was used to perform coarse grind simulations of wild-type CARD14 and two mutated structures. R547S increases protein flexibility, whereas V353I is shown to promote CARD14-induced NF-kappa B activation. This study confirms the association between two CARD14 nsSNPs, rs2066965 and rs34367357 with psoriasis in a Pakistani population, and could be helpful in identifying the role of CARD14 gene variants as potential genetic markers in patients with psoriasis.

## 1. Introduction

Psoriasis is a chronic inflammatory skin disease that manifests in red raised patches on the skin [1]. These patches may hamper the routine activities of patients. In the very recent past, psoriasis caught the attention of researchers due to its high prevalence and deleterious impact on human health and well-being. Psoriatic patches can appear at any time in the life span, but a high prevalence is observed after 50 years of age [2]. Studies have suggested that the prevalence of this disease is lower in developed countries compared to the developing world. In most developed countries, the prevalence of psoriasis is less than 5%. For instance, in the United States, approximately 2.2% of the population (7.5 million people) suffers from psoriasis [3], making it a prevalent autoimmune disease. In addition to skin-related complications associated with psoriasis, psoriatic arthritis causes painful joint inflammation and can affect other parts of the body. Studies have shown that the disease is equally prevalent in male and female populations [4].

A number of factors are associated with the susceptibility of psoriasis; however, 70% of psoriatic patients have been shown to have a family history [5]. This indicates that genetics play a very important role in the disease. The classical linkage analysis technique was first used to identify the genetic susceptibility regions for psoriasis in the human genome. This study revealed a total of nine regions in the human genome linked with the disease (known as Psoriasis susceptibility (PSORS)1–9) [6]. Among these regions, the PSORS-1, 2 and 4 were found to be linked in most of the genetic studies, indicating its important role in the genetic susceptibility of psoriasis [7,8,9]. The PSORS-1 locus is located on chromosome 6p21. Fine-mapping studies of this locus revealed a region of 150 kb associated with psoriasis. This region encompasses a total of nine genes. Among these nine genes, coding variations in three genes (HLA-C, CCHCR1, and CDSN) were found in the psoriatic patients [10]. PSORS-4 locus is on chromosome 1q21 and spans more than 60 genes. This region is also known as the epidermal differentiation cluster (EDC). A study revealed that the deletion of two genes (LCE3B and LCE3C) from this cluster resulted in psoriasis [9]. PSORS-2 locus maps on chromosome 17q25. Next-generation sequencing of two families revealed a mutation in one of the genes of this locus, the CARD14 (caspase recruitment domain-containing protein 14) gene [8].

The CARD14 gene in keratinocytes encodes an adaptor protein, which facilitates TRAF2 (TNF receptor-associated factor 2)-dependent NF-κB (nuclear factor kappa B) signal transduction [11]. So far, all known mutations in CARD14 have been shown to cause an activation of the NF-κB transcription factor [12,13]. This constitutive activation of NF-κB may lead to the overproduction of pro-inflammatory cytokines in keratinocytes. Interestingly, along with mutations, many single-nucleotide polymorphisms (SNPs) are found to be associated with psoriasis in case–control studies [11].

Despite a high prevalence of psoriasis in Pakistan, only a limited number of studies have been conducted to determine the genetics of this disease [14,15]. To our best knowledge, this is the first study in the Pakistani population to identify psoriasis-associated SNPs in CARD14. We utilized several bioinformatics tools, including molecular dynamic (MD) simulations with genotyping analysis to find the association of the most pathogenic variants of CARD14 gene and to determine possible impact of those variants on the structural stability of the CARD14 protein. We anticipate that this study will lay the foundation for designing better personalized therapeutics for patients with psoriasis in the future.

## 2. Results

### 2.1. Clinical Presentation of Psoriasis Patients

A total of 123 individuals from the same geographical area were recruited for this study after being informed about the purpose of the study and giving their formal written consent. Of these, 63 individuals were diagnosed with psoriasis, and the remaining 60 individuals served as controls. The mean age of the participants was 37 years (±14). The bodyweight of the participants was also recorded. The average body weight was 67 Kg (±16.38) in diseased individuals, while in healthy controls, the average body weight was 62.5 Kg (±10.45). The average waist circumference was 33 inches (±6.03) among the psoriatic patients; however, in healthy controls, the mean value of waist circumference was 32 inches (±2). Using participants’ weight and height values, BMI was calculated and the average BMI in psoriatic patients was 26.33 (±5.81), while in healthy individuals’, the average value was 24.3 (±2.3). We calculated a correlation between BMI and psoriasis in our dataset using the psych package in R. The correlation value was adjusted using FDR. Interestingly, a moderate positive correlation (r = 0.37) was found between BMI and PASI scores (Psoriasis Area and Severity Index: a scale to measure the severity of psoriasis in patients). The severity of psoriasis appeared to be higher in the BMI range 20–35 when compared to BMI < 20 and >35 (Figure 1a). It should be noted here that PASI scores of three patients were not included in our analyses due to unavailability.

Interestingly, an equal proportion of diseased males and females reported joint pain (Figure 1b), which is a major characteristic feature of psoriasis [16]. PASI scores were categorized as mild, moderate and, severe [17]. Interestingly, in our data, a relatively higher number of male patients reported high PASI scores (>15) (Figure 1c). Furthermore, the dataset also indicated the seasonal influence on the disease and 70% of individuals reported the aggravation of the disease condition during cold weather. In addition, 70% of the individuals reported a family history of psoriasis (data not shown).

### 2.2. Identification and Selection of nsSNPs

The CARD14 SNPs dataset was downloaded from the Ensemble genome browser. We focused on the missense or nonsynonymous SNPs (nsSNPs) due to their potential impact on the protein structure. A total of 7311 nsSNPs were found in the gene. To further shortlist the number of nsSNPs for this study, we applied several computational tools to organize deleterious nsSNPs that may disturb the CARD14 protein structure, and therefore be associated with the disease. A total of six algorithms were applied to sort the deleterious nsSNPs, including SIFT, Polyphen2, Reval, CADD, MetaLR and mutation assessor [18,19,20,21,22,23]. SIFT mainly classified the nsSNPs into two different classes: deleterious and tolerated. SIFT identified 3331 nsSNPs that were classified as deleterious, while the remaining nsSNPs were classified as either tolerated or less deleterious (Figure 2a). Polyphen2 predicts the impact of nsSNPs on proteins’ structure and function using the evolutionary comparative analysis [19]. This tool classifies the substitutions in three different categories: probably damaging, possibly damaging, and benign. In the CARD14 gene, 2984 nsSNPs were categorized as probably and possibly damaging by Polyphen2 (Figure 2b). Similarly, other bioinformatics tools such as CADD, MetaLR, and Revel sorted the nsSNPs into deleterious and benign. These three tools predicted a smaller number of substitutions as deleterious compared to SIFT and Polyphen2. A total of 61, 159, and 16 nsSNPs were predicted to be disease-associated by CADD (Figure 2c), Revel (Figure 2d), and MetaLR (Figure 2e), respectively. Similarly, the mutation assessor also predicted only 24 nsSNPs that could be highly damaging (Figure 2f). Among them, only four nsSNPs that were predicted to be deleterious by combining all tools were selected for this study. Furthermore, fourteen previously associated nsSNPs in CARD14 gene (Table 1) were included in the genotyping [24].

### 2.3. Genotyping of CARD14 in Diseased and Control Samples

Sanger sequencing was used to sequence the different regions of the CARD14 gene encompassing our selected nsSNPs. Sequencing data were analyzed to identify the presence or absence of the nucleotides at selected positions in diseased and healthy control samples. Based on the observed frequencies of the alleles in the samples, odds ratios were calculated at 95% significance to determine the association of SNPs with psoriasis. Interestingly, between eighteen putative SNPs, only two nsSNPs, rs2066964 and rs34367357, were found to be linked with psoriasis in the Pakistani population. Furthermore, rs2066964 (R547S) and rs34367357 (V585I) nsSNPs showed odd ratio values 1.01 and 1.49, respectively. Interestingly, none of the novel nsSNPs identified using computational methods was found to be linked with psoriasis in our study population.

We further determined the association between rs2066964 and rs34367357 with other factors present in our dataset, including BMI, psoriasis severity, and joint pain. Interestingly, it was found that most psoriatic patients from all categories of BMI were heterozygous. Similarly, the G homozygous genotype was also present in all BMI categories except underweight patients (Figure 3a). In contrast, a homozygous G genotype was absent in most of the diseased individuals in the case of rs34367357 (Figure 3b). This indicates the possibility of a potential masking effect of heterozygosity in rs34367357, which may protect the individuals from the disease [25]. In addition, the homozygosity and heterozygosity distribution of both SNPs was also observed in psoriatic patients with and without joint pain. In the case of rs2066964, heterozygous genotype was highly prevalent in the patients with joint pain, while the rs34367357 G-homozygous was found to be common in the patients with pains (Figure 3c,d). A similar type of genotype distribution was also found with reference to PASI score (Figure 3e,f).

### 2.4. Structural Analysis of SNPs Associated with Psoriasis

We further performed bioinformatics analysis of associated nsSNPs with the disease to gain an insight into molecular mechanisms. The 3D structure of CARD14 was predicted using the state-of-the-art protein structure prediction tool alpha-fold2. This tool was employed due to its high prediction accuracy compared to other contemporary tools [26]. After 3D structure prediction, the InterPro server was used to predict the domains in the CARD14 protein structure [27]. InterPro predicted five different domains in the protein structure, including CARD domain, coiled-coil domain, inhibitory domain, PDZ domain, and guanylate kinase-like domain. Additionally, a coiled-coil domain and inhibitory domain are found in CARD14 (Figure 4a). The associated nsSNP rs2066965 (R547S) is in the inhibitory domain of CARD14, while rs34367357 (V585I) is present in the PDZ domain (Figure 4b). It can be speculated that the substitution of R547 with serine in the inhibitory domain results in the constitutive activation of CARD14, which may lead to its dimerization with NF-kB to trigger the inflammation in keratinocytes. Similarly, the substitution in PDZ domain (V585I) results in the increased dimerization of NF-kB. This enhanced dimerization may be a potential trigger for the disease.

We further used NMsim (normal mode-based geometric simulations) webserver to develop an idea about the dynamic impact of substitutions in the protein structure. Root mean square deviation (RMSD), root mean square fluctuation (RMSF), the radius of gyration (Rg), and solvent accessible surface area (SASA) of the wild-type and mutated structures were calculated. The wild-type and R547S showed slightly higher RMSD values compared to the V585I (Figure 5a). This indicates that R547 substitution with serine in the inhibitory domain does not affect the conformation of the protein. Interestingly, R547S substitution led to increased flexibility in the protein structure compared to the wild-type and V585I as shown in the RMSF plot (Figure 5b). This increased flexibility may lead to a higher activation of CARD14 protein. Surprisingly, the values of radius of gyration (Rg) and solvent accessible surface area (SASA) were higher for both mutants as compared to wild-type. This highlights that both mutants are maintaining their open confirmation to be active (Figure 5c,d).

## 3. Discussion

Psoriasis is a chronic, inflammatory skin disease that is caused by immune responses. It is a debilitating condition that negatively affects quality of life and even life expectancy. Several studies have indicated the role of genetics in the manifestation of the disease. CARD14 is a scaffold protein and is abundantly found in the skin. Its role in cell polarity, cell–cell adhesion, and signal transduction is well-documented [28,29]. Studies indicate an association between CARD14 gene expression dysregulation and polymorphism with psoriasis [30,31,32]. The prevalence of psoriasis in the Pakistani population is high; however, very few studies have been conducted to evaluate the CARD14 genetic polymorphism contribution in the development of this disease [14,15]. Therefore, the aim of the current study was to delineate the possible association of nsSNPs of CARD14 gene in the Punjabi population of Pakistan. The study coupled several bioinformatics tools, including MD simulations with genotyping analysis to find an association of the most pathogenic variants of CARD14 with the Pakistani population and to determine the possible impact of those variants on the structural stability of the CARD14 proteins.

Previously, the association of CARD14 variants with psoriasis was reported in Japanese [33], Spanish [34], European [35], Chinese [31,32,36,37], and Tunisian [38] populations. Mechanistically, the mutations in the CARD14 gene lead to hyperinflammation in the keratinocytes, which causes immune cell infiltration, hyperproliferation of keratinocytes, and keratosis, which are typical hallmarks of psoriasis [39]. An in vivo study by Mellett et al. [40] provided significant insights into the disease mechanism driven by CARD14 gain of function mutations. They introduced Card14ΔE138K/A mutations in mice with psoriasis and found an up-regulation of pro-inflammatory cytokines, IL-23, IL-19, IL-22 and IL-17; chemokines Cxcl1, Ccl20, and Cxcl2; and antimicrobial peptides, Lcn2 and Defb4. Their study further indicated that CARD14 E138 mutations enhance the proteolytic cleavage-dependent activation of MALT1 and derive the formation of the CBM complex. The role of E138A mutation in inducing psoriasis pathogenicity by up-regulating IL-17A has been reported [30]. Furthermore, the up-regulation of pro-inflammatory chemokines and cytokines has been shown in a Taiwanese family with a familial psoriasis [12].

In our study, we delineated the most pathogenic nsSNPs of the CARD14 gene using six bioinformatics tools (SIFT, PolyPhen, REVEL, CADD, MetaLR, and Mutation Assessor). These tools employ different criteria based on protein sequence homology and conservation, amino acid structural role, and protein function to classify variants as disease-causing/deleterious/damaging. Based on the pathogenicity score of each tool, we identified four nsSNPs from CARD14 as deleterious. Furthermore, we used a high-throughput Sanger sequencing method to find the association between the predicted four variants along with the fourteen previously reported CARD14 SNPs with the disease. None of the predicted nsSNPs was associated with the disease in our study population. These variants should be tested in other populations for their association with psoriasis. Furthermore, the algorithms that each tool uses to make predictions need to be updated to enhance the prediction accuracy of the tools.

Two previously reported nsSNPs, rs2066964 and rs34367357, were found to be associated with psoriasis in a Pakistani population. In Hungarian patients, the association of rs2066964, as well as rs117918077, rs28674001, and rs11652075, was demonstrated with an up-regulation in the NF-kB pathway [41]. Both rs34367357 and rs2066964 were also associated with pustular psoriasis and psoriasis vulgaris in a Hans Chinese population [42]. These two nsSNPs were also reported to be associated with pityriasis rubra pilaris in a Hungarian population [43]. The association of nsSNPs rs34367357 and rs2066964 with disease severity, joint pain, and BMI was also determined. We found that an rs2066964 GG genotype was present in all BMI groups other than underweight individuals. Similarly, the high frequency of rs2066964 heterozygous genotypes was observed in patients with joint pains. Similar to our study, a recent study also reported no association of SNP rs34367357 with joint pain in psoriasis [44]. Interestingly, in our study population, we found a moderate positive correlation between the severity of psoriasis and BMI, although the PASI score appeared to be higher in the BMI range 20–35, when compared to BMI above 35. Earlier published studies (reviewed in [45]), suggest a significant positive association between psoriasis severity and BMI with some exceptions where no significant association was observed between BMI and the PASI score [46]. Thus, further detailed studies are required to determine if obesity is the consequence of psoriasis or a risk factor for this skin disease.

Furthermore, CARD14 protein consists of five domains: CARD domain, coiled-coil domain, inhibitory domain, PDZ domain, and guanylate kinase-like domain. Among these domains, the CARD domain (caspase recruitment domain) is the most important domain in this protein. This domain is made up of a bundle of six alpha-helices and plays a critical role in inflammation by interacting with NF-kB [47]. This process involves assembling multi-protein complexes to facilitate the dimerization of CARD14 with NF-kB, and thus its activation. Likewise, the PDZ domain present in CARD14 helps with the interaction carboxyl-terminal of the target protein [48]. Guanylate kinase-like domain of the CARD14 is involved in catalysis of the ATP-dependent phosphorylation of GMP to GDP [49].

Two nsSNPs, rs2066965 and rs34367357, associated with psoriasis in our study population were present in the inhibitory domain and PDZ domain, respectively. The substitution of R547S might lead to the constitutive activation of the protein, whereas V535I substitution might enhance dimerization of NF-kB, suggesting the possible role of CARD14 in the disease. A protein structure analysis, however, revealed that R547S substitution did not affect protein structure but enhanced the flexibility of the protein. Studies suggest that this increase in flexibility allows proteins to make more stable molecular interactions with other molecular partners. Therefore, it can be speculated that R547S substitution-mediated protein flexibility allows CARD14 to make stronger interactions with its down-stream targets. However, further experiments are necessary to validate this hypothesis.

## 4. Materials and Methods

### 4.1. Study Subjects and Sample Collection

This study protocol was reviewed and approved by the Board of Advanced Studies and Research (BASR) of the University of Lahore, Lahore, Pakistan, (approval number 57/Item 28; 47th BASR meeting; 5 November 2020). All participants of the study were properly informed about the purpose of the study and were asked to complete the designed questionnaire. A written informed consent was obtained, and the objective of research study and method of blood sampling were explained to the patients and control individuals in detail. Detailed history and examination of patients were performed. Phenotypic variations were examined, and lesions were evaluated for thickness, scaling, erythema, and body area involvement. The severity of the disease was evaluated using PASI score. The blood pressure, BMI, and waist circumference of the subjects were also recorded. Using sterile techniques, blood samples were collected in CBC vials with/without EDTA as an anticoagulant. Samples were collected from patients of Aziz Bhatti Shaheed Hospital, Gujrat and various private skin clinics of the Gujranwala division, District Headquarter Hospital Faisalabad, and Sheikh Zaid hospital Rahim Yar Khan. Control samples were obtained from healthy individuals living in the same locality as patients. DNA was extracted from the blood for genotyping of nsSNPs in CARD14 gene.

### 4.2. Selection of nsSNPs

Ensemble genome browser was used to download the SNPs data of the CARD14 gene, which contained all types of SNPs, including 3’-UTR, 5’-UTR, non-coding, missense, nonsense, etc. The data were filtered and nsSNPs (missenses) were selected for the study due to their ability to substitute amino acids in protein sequences, which may impact the structural stability of proteins and contribute to disease susceptibility [50]. The nsSNPs were sorted for their potential pathogenic impact on the protein structure using various bioinformatics tools, including PolyPhen-2, SIFT, CADD, REVEL, MetaLR, and Mutation accessor [18,19,20,21,22,23]. These tools classified the nsSNPs as pathogenic, moderately pathogenic, and non-pathogenic. The SNPs, which are predicted to be deleterious by all tools, were selected for analysis. Along with the 4 predicted nsSNPs, 14 SNPs already reported in the literature [24] were genotyped.

### 4.3. PCR Amplification and Sanger Sequencing

Sanger sequencing was used to sequence the regions of the CARD14 gene, harboring the nsSNPs in disease and healthy control samples [38]. The primers were designed using the Primer-3 tool and gene regions were amplified using the optimum Tm value (Appendix A) [51]. The PCR products were purified, and sequencing was performed at the University of Wisconsin—Madison Biotechnology Center’s DNA Sequencing Facility (Research Resource Identifier—RRID: SCR_017759). Later on, sequencing data were analyzed for each SNP to determine its prevalence in the population.

### 4.4. Statistical Analysis

To calculate the association of the nsSNPs with the CARD14 gene, the odds ratio of each SNP was calculated using a 95% confidence interval. An odds ratio > 1 indicates the positive association of SNPs with the disease [52].

### 4.5. 3D Structure Prediction and Simulations

We performed a detailed in silico analysis to determine the impact of disease-associated nsSNPs on the CARD14 protein structure. The current state-of-the-art protein 3D structure prediction tool, alpha-fold was employed to predict the 3D structure of the protein [26,53]. Currently, an experimental 3D structure of human CARD14 is not available in protein databank. InterPro was used to identify the domains in the protein structure [27]. The amino acids substitutions in the CARD14 were carried out using Pymol. Finally, NMSIM server was used to perform coarse-grind simulations of wild-type CARD14 and two mutated structures to determine the impact of mutations at dynamic level. After successful completion of the simulations, wild-type and mutated trajectories were used to calculate root mean square deviation (RMSD), root means square fluctuation (RMSF), the radius of gyration (Rg), solvent accessible surface area (SASA) to understand the dynamic behavior of proteins.

## 5. Conclusions

Psoriasis is a disease condition that leads to severe physical, social, and psychological consequences for the individual suffering from it. In this study, for the first time, we reported the association of two nsSNPs, rs2066965 and rs34367357, in the CARD14 gene with psoriasis in Pakistani population. nsSNP rs2066965 heterozygous genotype is more prevalent in patients with joint pain. However, further experimental validation is required to find the SNP association with joint pain and to understand the possible molecular mechanism that leads to joint pain in rs2066965 heterozygous individuals. Simulation analysis provided an insight behind the molecular basis of the disease. R547S increases protein flexibility, whereas V353I could promote CARD14-induced NF-kB activation. Similar studies must be performed in other populations to further delineate the association and exact role of rs2066965 and rs34367357 in psoriasis. Moreover, in vitro mutagenic studies could be conducted to better understand the role of these nsSNPs in modulating CARD14 activity and protein interactions. The outcomes of this study are a step forward in identifying the role of CARD14 gene variants as potential genetic markers.

## Figures and Tables

**Figure 1 ijms-23-09336-f001:**
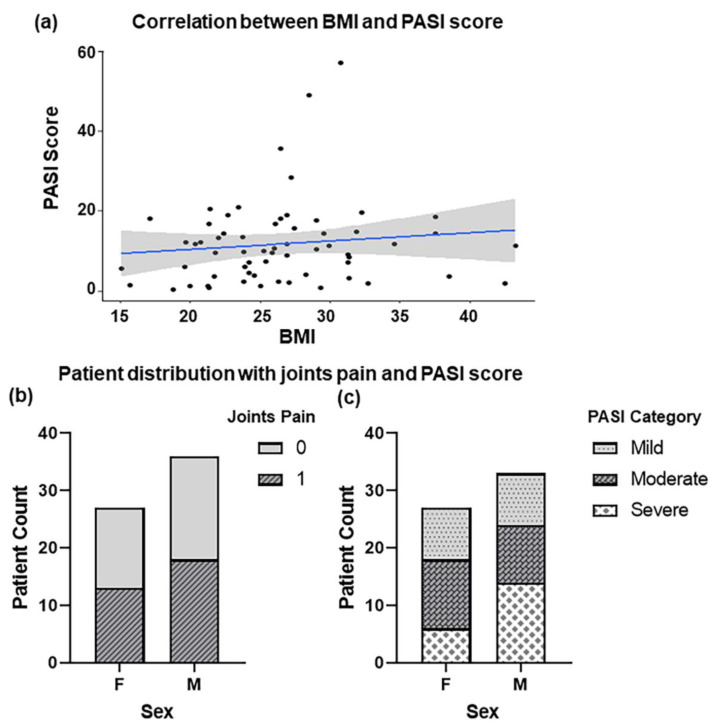
Clinical presentation of psoriasis patients. (**a**) A positive correlation of 0.37 between BMI and PASI score of the patients was observed. (**b**) The proportion of male and female patients that reported joint pain, (**c**) Bar chart showing the severity of psoriasis (PASI scores) in both genders.

**Figure 2 ijms-23-09336-f002:**
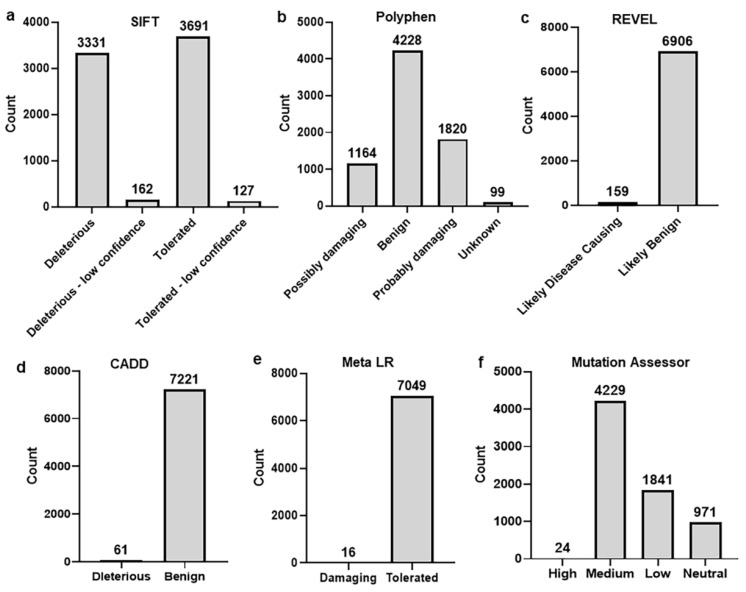
Sorting of nsSNPs using six different algorithms: (**a**) SIFT, (**b**) Polyphen2, (**c**) Reval, (**d**) CADD, (**e**) Meta LR, (**f**) Mutation assessor.

**Figure 3 ijms-23-09336-f003:**
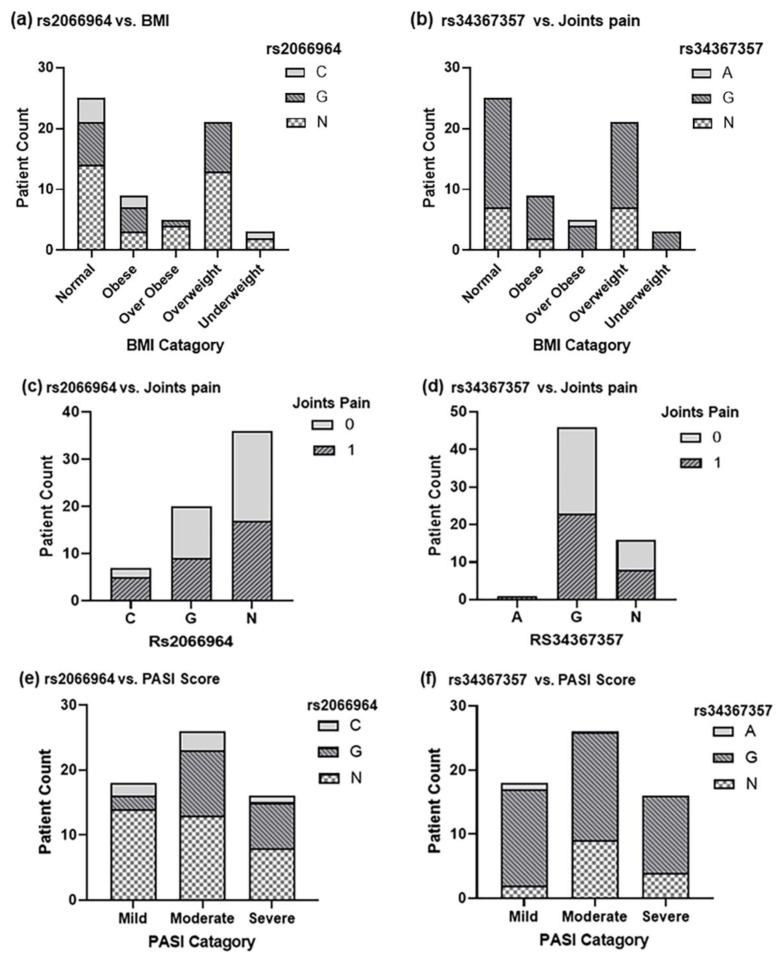
Distribution of different genotypes in the psoriatic patients. (**a**) Bar chart shows the distribution of rs2066964 genotypes in the psoriatic patients in different BMI categories. (**b**) Bar chart shows the distribution of rs34367357 genotypes in the psoriatic patients in different BMI categories. (**c**) Bar chart shows the distribution of rs2066964 genotypes in the psoriatic patients with and without joint pains. (**d**) Bar chart shows the distribution of rs34367357 genotypes in the psoriatic patients with and without joint pains. (**e**) Bar chart shows the distribution of rs2066964 genotypes in the psoriatic patients in different PASI score categories. (**f**) Bar chart shows the distribution of rs34367357 genotypes in the psoriatic patients in different PASI score categories.

**Figure 4 ijms-23-09336-f004:**
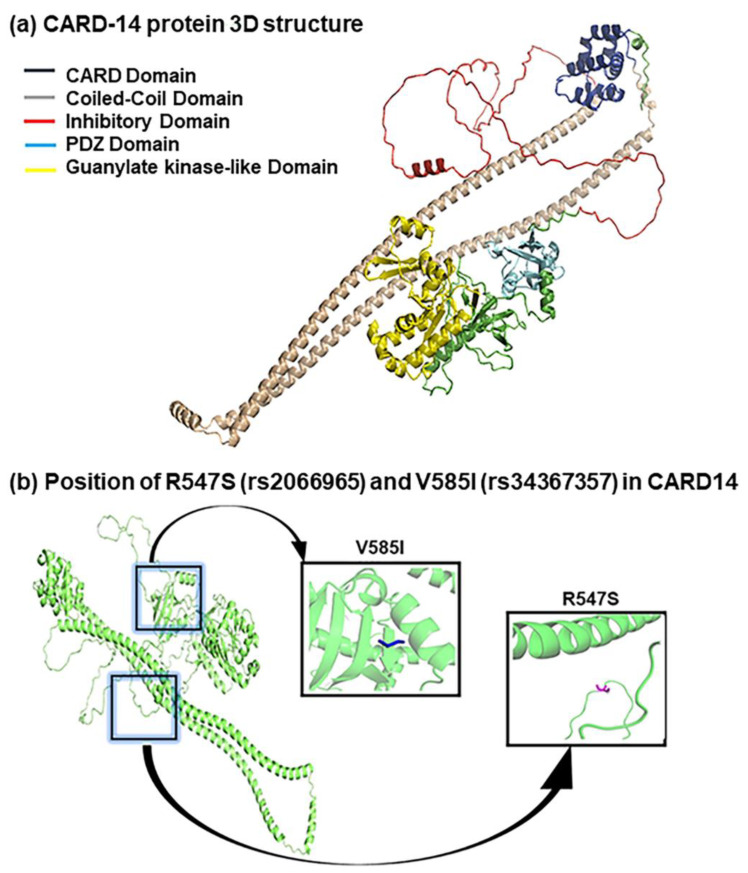
Structural analysis of CARD14 protein. (**a**) CARD14 protein 3D structure predicted using Alpha-Fold2. InterPro predicted five domains in the CARD14 structure, including CARD domain (Blue), coiled-coil domain (Gray), inhibitory domain (Red), PDZ domain (Cyan), and guanlate kinase-like domain (Yellow). (**b**) Position of R547S (rs2066965) and V585I (rs34367357) in CARD14 3D structure. The R547S is located in the inhibitory domain of CARD14, while V585I is present in the PDZ domain.

**Figure 5 ijms-23-09336-f005:**
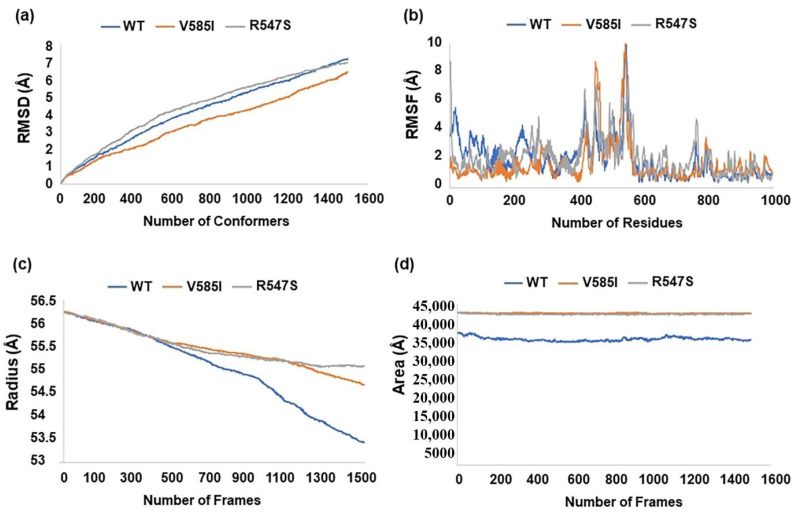
Dynamic impact of substitutions in the CARD14 protein structure. (**a**) Plot of root mean square deviation (RMSD). (**b**) Plot of root mean square fluctuation (RMSF). (**c**) Plot of radius of gyration (Rg). (**d**) Plot of solvent surface accessible area (SASA).

**Table 1 ijms-23-09336-t001:** List of CARD14 nsSNPs genotyped in this study.

Sr. No.	nsSNPs ID	Alleles	Amino Acid
1	rs146214639	T/G	L150R
2	rs1598639617	T/C	L124P
3	rs1598639659	G/C	C127S
4	rs1598639974	A/C	Q157P
5	rs281875215	G/A	G117S
6	rs1567872320	G/A	E138K
7	rs281875212	G/A	E142K
8	rs281875213	A/G	E142G
9	rs387907240	T/C	L156P
10	rs281875216	C/A	H171N
11	rs2066964	G/C	R547S
12	rs34367357	G/A	V585I
13	rs200102454	C/T	T591M
14	rs281875220	T/A	I593N
15	rs201285077	C/T	S602L
16	rs117918077	C/T	R682W
17	rs1567903243	C/A	S802R
18	rs11652075	C/T	R820W

## Data Availability

The datasets generated for this study are available on request to the corresponding authors.

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
