# Peer review of "Association of CARD14 Single-Nucleotide Polymorphisms with Psoriasis"

_ijms, 2022, doi:10.3390/ijms23169336_

Round 1

Reviewer 1 Report

In this manuscript, the authors analyzed the correlations of severity of psoriasis with BMI, sex, and genotypes of human subjects with or without psoriasis.  The distribution and impact of these mutations associated with psoriasis in patients were also analyzed.  The study could lead to further elucidate the mechanisms that cause the disease and develop therapeutics for psoriasis.  The manuscript is clearly written.  One minor concern is that the authors claim that the severity of psoriasis is positively correlates with BMI (Fig. 1a).  The statistics has no problem.  However, the figure really shows that the disease appears to be in the region of BMI 20-35.  When BMI is above 35, the severity seems reduced.  The authors might want to clarify this.

Author Response

Response to Reviewer 1 Comment

Point 1: One minor concern is that the authors claim that the severity of psoriasis is positively correlates with BMI (Fig. 1a).  The statistics has no problem.  However, the figure really shows that the disease appears to be in the region of BMI 20-35.  When BMI is above 35, the severity seems reduced.  The authors might want to clarify this.

Response 1: We would like to thank the reviewer for their thorough reading of our manuscript. We agree with the reviewer on this point that in our study population, psoriasis severity appears to be high in the region of BMI 20-35, compared to BMI above 35. We have clarified this in the result section (page 3, line 99-100), and in the discussion section (page 9, line 266-272) including two additional references (45-46) in the revised manuscript. The changes are highlighted in yellow. We hope this will suffice the reviewers’ minor concern.

Reviewer 2 Report

The paper describes the investigation of potential association of CARD14 single nucleotide polymorphisms with psoriasis. The authors have designed a scientifically sound experiment which revealed for the first time the association of two nsSNPs, rs2066965 and rs34367357 in CARD14 gene with psoriasis in Pakistani population, findings that might orientate future research towards the identification of the role of CARD14 gene variants as potential genetic marker for psoriasis. The methods were clearly described thus being entirely replicable and provide plausible results. The findings were presented in a logical succession and further discussed against previously published data. The article is fluently presented in an academic manner.

Author Response

Response to Reviewer 2 Comments:

We would like to thank the reviewer for their thorough reading of our manuscript and appreciating our study. There are no additional comments to respond.